# Adipose-Derived Stromal Cells Exposed to RGD Motifs Enter an Angiogenic Stage Regulating Endothelial Cells

**DOI:** 10.3390/ijms26030867

**Published:** 2025-01-21

**Authors:** Nicolo-Constantino Brembilla, Sanae El-Harane, Stéphane Durual, Karl-Heinz Krause, Olivier Preynat-Seauve

**Affiliations:** 1Hekestiss Plan-les-Ouates, 1228 Geneva, Switzerland; 2Department of Medicine, Faculty of Medicine, University of Geneva, 1206 Geneva, Switzerland; sanae.elharane@unige.ch; 3Department of Pathology and Immunology, Faculty of Medicine, University of Geneva, 1206 Geneva, Switzerland; karl-heinz.krause@unige.ch; 4Laboratory of Biomaterials, Faculty of Dental Medicine, University of Geneva, 1206 Geneva, Switzerland; stephane.durual@unige.ch

**Keywords:** adipose-derived stromal cells, RGD, angiogenesis

## Abstract

Adipose-derived stromal cells (ASCs) possess significant regenerative potential, playing a key role in tissue repair and angiogenesis. During wound healing, ASC interacts with the extracellular matrix by recognizing arginylglycylaspartic acid (RGD) motifs, which are crucial for mediating these functions. This study investigates how RGD exposure influences ASC behavior, with a focus on angiogenesis. To mimic the wound-healing environment, ASC were cultured in a porcine gelatin sponge, an RGD-exposing matrix. Transcriptomics revealed that ASC cultured in gelatin exhibited an upregulated expression of genes associated with inflammation, angiogenesis, and tissue repair compared to ASC in suspension. Pro-inflammatory and pro-angiogenic factors, including IL-1, IL-6, IL-8, and VEGF, were significantly elevated. Functional assays further demonstrated that ASC-conditioned media enhanced endothelial cell migration, tubulogenesis, and reduced endothelial permeability, all critical processes in angiogenesis. Notably, ASC-conditioned media also promoted vasculogenesis in human vascular organoids. The inhibition of ASC-RGD interactions using the cyclic peptide cilengitide reversed these effects, underscoring the essential role of RGD-integrin interactions in ASC-mediated angiogenesis. These findings suggest that gelatin sponges enhance ASC’s regenerative and angiogenic properties via RGD-dependent mechanisms, offering promising therapeutic potential for tissue repair and vascular regeneration. Understanding how RGD modulates ASC behavior provides valuable insights into advancing cell-based regenerative therapies.

## 1. Introduction

Adipose-derived Stromal cells (ASCs) are multipotent stem cells capable of differentiating into various cell types, including adipocytes, osteocytes, and chondrocytes. They are easily isolated from adipose tissue and produce a regenerative secretome, making them a convenient tool for cell-based regenerative therapies [1].

Arginylglycylaspartic acid (RGD) is a key peptide motif involved in cell adhesion to the ExtraCellular Matrix (ECM). In mature, steady-state tissues, which are primarily composed of collagens and laminins, RGD motifs are embedded within cryptic domains, making them inaccessible. Consequently, limited stimulation through RGD causes endothelial cells on the inner surface of vessels to exhibit low levels of proliferation and migration. During healing, the release of Matrix MetalloProteinases (MMPs) and the formation of a provisional matrix enriched with RGD-exposing proteins, such as fibronectin, fibrinogen, and vitronectin, activate endothelial cells to initiate angiogenesis. The release of angiogenic factors further shifts the integrin expression of endothelial cells towards RGD-binding integrins, thereby amplifying tissue repair [2]. ASC expresses several RGD-binding integrins that play critical roles in their interactions with the extracellular matrix (ECM) components and in mediating cell behavior. Among the most prominent RGD-binding integrins expressed by ASC are αvβ3, αvβ5, and α5β1. These integrins recognize RGD motifs found in ECM proteins such as fibronectin, vitronectin, and denatured collagens, facilitating processes like adhesion, migration, and signaling pathways involved in angiogenesis, survival, and differentiation.

During healing, ASCs are mobilized from adipose tissue to the injury site, where they play a crucial role in promoting regeneration and angiogenesis. As the wound environment shifts towards a provisional matrix enriched with RGD, the recruited ASC are exposed to high levels of RGD. Of note, ASC expresses various integrins, including those that specifically bind to RGD [3,4]. Little is known about the impact of exposure to RGD-containing motifs on the regenerative properties of ASC. It is known that fibronectin promotes their proliferative and migratory properties [5,6] leading to their mobilization to the healing site. The RGD-binding integrin α5β1 has been shown to be involved in the adhesion of ASC to the ECM [7] and their migration to angiogenic sites for differentiation into endothelial cells is mediated by α5β1 [8,9]. Furthermore, the integrin-mediated adhesion of ASC to RGD has been demonstrated to influence the differentiation potential of ASC, directing them towards specific lineages depending on the ECM composition [10]. For instance, the use of RGD-functionalized peptide hydrogels has been shown to stimulate growth factor secretion in human amniotic mesenchymal stem cells, enhancing their therapeutic effects in wound healing [10]. Additionally, integrin αvβ3 has been implicated in the early osteogenic differentiation of mesenchymal stem cells [11]. Other studies have shown that RGD-containing peptides can enhance the endothelial differentiation of ASC by activating focal adhesion kinases and the PI3K/Akt pathway, leading to the increased expression of CD31 and von Willebrand factor [12]. Also, indirect observations based on the interaction between gelatin and mesenchymal stromal cells, including ASC, suggested the biological activities of RGD on ASC. Indeed, gelatin is an irreversibly heated/denatured form of collagen, exposing the needed spatial conformation of RGD motifs to RGD-binding integrins [13,14]. In general, the RGD motifs of gelatin ligate RGD-binding integrins in the presence of divalent cations [15]. Some studies have reported the biological activity of gelatin on ASC, indirectly suggesting a biological role of RGD: mesenchymal stem cells loaded with gelatin microgels have been shown to increase HGF secretion and their anti-fibrotic effects on altered kidneys [16]. The introduction of mesenchymal stem cells into gelatin increased the secretion of VEGF, FGF-7 (KGF), and HGF [15]. Furthermore, the exposure of ASC to a thick layer of cross-linked gelatin promoted their proliferation [17]. Recent work demonstrated that ASC cultured in a gelatin sponge regulates their angiogenic capacities, both in vitro and in vivo [18]. Together, these observations suggest a novel regulatory mechanism of ASC in response to a healing environment that exposes RGD motifs to cells, highlighting the importance of the microenvironment in cellular functions. Understanding the regulation of ASC through their interaction with exposed RGD motifs, a situation occurring during the healing process, is expected to have implications for understanding their regenerative, and notably angiogenic, capacities for therapeutic purposes.

In this study, we used a gelatin sponge-based method to expose ASC to RGD and investigated the impact of RGD exposure on ASC functions. The key factors influencing this choice are as follows: (i) their ability to deliver RGD motifs to cells more effectively than native collagen; (ii) their capacity to mimic the denatured collagen environment typically present in wound healing; (iii) their favorable physicochemical properties, such as porosity and flexibility, which facilitate the introduction of cells and culture medium; and (iv) their compatibility with clinical applications, including a lack of toxicity, excellent biocompatibility, and bioresorbability that make them suitable for use in humans.

## 2. Results

### 2.1. A Porcine Gelatin Sponge Regulates the Transcriptome of ASC in Favor of a Biological Response Involving Pro-Inflammatory/Angiogenic Factors

Gelatin is an irreversibly heated/denatured form of collagen, exposing the needed spatial conformation of RGD motifs to cells [13,14]. A porcine gelatin sponge was previously used as a matrix scaffold for ASC culture [18]. As previously described [18], ASC compaction and survival within the gelatin trabeculae were shown (Figure 1A). A Principal Component Analysis of the transcriptomes from three independent ASC lines indicated that introduction into gelatin sponge scaffolds induced significant changes in gene expression compared to ASC in suspension from a monolayer culture, or ASC clustered in spheroids or ASC in 3D without a matrix (Figure 1B). All the conditions demonstrated distinct regulatory patterns, suggesting that introduction within a gelatin sponge influences ASC towards a specific behavior. According to the gene analysis and annotation resource Metascape (metascape.org), the most regulated genes between ASC/gelatin sponge and ASC in suspension from a monolayer culture (fold change > 5 and <5, *p* < 0.05) showed attribution to functions linked to wounding, tube morphogenesis, circulatory system process, response to growth factors, and inflammation (Figure 1C). A targeted analysis of the factors involved in the wound-healing process confirmed the upregulation of cytokines and chemokines with inflammatory/angiogenic properties (Figure 2). Among these factors, the most significant upregulations concerned the pro-inflammatory/angiogenic factors *IL-8* (*CXCL-8*), *IL-6*, and *IL-1*. Angiogenic *angiopoietins*, *HGF*, *CXCL-1*, and *CXCL-5 genes* were also upregulated. The metalloproteinases *MMP3* and *MMP-8* genes, involved in the angiogenic process [19], were also increased. Of note, the time course of angiogenic activation in ASC exposed to a gelatin sponge, analyzing the transcriptomes at 24 h, 48 h, 4 days, and 8 days, showed that angiogenic regulation was established after 48 h and subsequently maintained without significant changes between 48 h and 8 days (not shown). A Gene Set Enrichment Analysis (GSEA) was also performed to assess whether some genes were overrepresented in the most regulated functional families, helping to identify a top list of genes of particular importance in these regulations. The pro-inflammatory/angiogenic factors *IL-1*, *IL-6*, *IL-8*, *CCL-2* (*MCP1*), and *VEGF* were top-ranked (Figure 3), confirming gene regulation linked to inflammation and angiogenesis when ASC is exposed to a gelatin sponge. A secretome mass spectrometry analysis of three independent ASC lines exposed to a gelatin sponge was also compared with the same ASC lines in suspension from a monolayer culture. Hierarchical clustering established a clear separation between the two conditions, confirming the impact of ASC exposure to gelatin on their secretions (Figure 4A). The functional attribution of the most regulated proteins (fold change > 5 and <5, *p* < 0.05) confirmed the impact of gelatin on the following functions: the development of vasculature, blood circulation, and the positive regulation of angiogenesis (Figure 4B). Thus, the exposure of ASCs to a gelatin sponge induces gene regulation in favor of the secretion of the factors involved in the angiogenic process.

### 2.2. A Porcine Gelatin Sponge Regulates the Secretome of ASC in Favor of Regulations of Endothelial Cell Functions

Angiogenic factors act primarily on endothelial cell proliferation, migration, tubular assembly, and permeability to promote angiogenesis. In vitro functional assays were performed to assess the impact of the gelatin sponge on the ability of ASC to influence endothelial cell functions. To study the ASC secretome impact on endothelial cells, conditioned media were prepared using the following conditions: ASC/gelatin sponge, the same number of ASC without gelatin, and gelatin sponge alone. The experimental setup was the following: ASC in gelatin sponge was used to condition a HUVEC medium for three days. While RGD motifs were primarily exposed to ASC by the “solid” gelatin sponge, we could not entirely rule out the possibility that some RGD-exposing gelatin fibers might become solubilized in the conditioned media and could directly influence endothelial cell behavior independently of ASC. Thus, we included control conditions using empty gelatin sponges to account for the potential presence of soluble gelatin fibers exposing RGD and directly interfering with endothelial functions. Human umbilical vein endothelial cells (HUVECs) were then cultured in the conditioned media to assess the impact on their proliferation, migration to a fibronectin layer, tubular assembly, and permeability.

The medium conditioned by ASC/gelatin did not induce any changes in the growth of HUVECs compared to the same number of ASC compacted without gelatin (Figure 5A), indicating no impact on cell proliferation. As expected, the absence of ASC (medium conditioned by an empty gelatin sponge) significantly reduced the growth of HUVEC, consistent with the basal production of endothelial cell growth factors by ASC. The same conditioned media were evaluated for their impact on the ability of HUVEC to migrate through a layer of fibronectin. Compared to the medium conditioned by ASC alone or by an empty gelatin sponge, the medium conditioned by ASC in a gelatin sponge increased their capacity to migrate through a layer of fibronectin (Figure 5B). Compared to the medium conditioned by ASC alone or by an empty gelatin sponge, the medium conditioned by ASC in a gelatin sponge significantly enhanced their capacity to form tubes in appropriate culture conditions (Figure 5C,D). During the angiogenic process and wound healing, endothelial cell permeability is also an important function that undergoes dynamic changes, with both increases and decreases occurring at different stages to either favor inflammatory cell migration (increased permeability) or stability of vessels (decreased permeability). The impact of the differently conditioned media on HUVEC permeability was also studied. A compact monolayer of HUVEC was established and exposed to the conditioned media before measuring its permeability to a fluorescent particle (FITC-dextran). TNF-α, used as an inducer of endothelial permeability, increased permeability to FITC-dextran (Figure 5E). The medium conditioned by an empty gelatin sponge did not induce any change. However, the media conditioned by ASC alone or ASC in a gelatin sponge strongly decreased endothelial permeability in favor of vessel stabilization. Together, these data indicate that ASCs exposed to gelatin, compared to the same number of ASCs compacted without gelatin, do not change their ability to promote endothelial proliferation but enhance their migration and tubulogenic capacities.

### 2.3. RGD-Binding Participates in ASC Attachment to Gelatin

The ability of ASCs to bind to porcine gelatin was investigated. ASC plated onto porcine gelatin gels (6%) induced rapid gel dissolution, probably due to the secretion of MMPs with gelatinase activity (not shown). Thus, gelatin gel cross-linking was necessary to preserve gelatin integrity. Compared with the ASC plated on plastic, the ASC plated on cross-linked porcine gelatin gel was denser, favoring cell proliferation (not shown). ASCs were also seeded on gelatin gel in the presence of Cilengitide, a cyclic RGD-containing peptide inhibiting binding to RGD motifs [20]. Cilengitide prevented ASC attachment to gelatin (Figure 6A) but maintained the cells alive without inducing cell death (Figure 6B). Thus, RGD motifs are suggested to be involved in the ASC/gelatin interaction and could participate in the angiogenic regulations.

### 2.4. RGD-Binding Is Involved in the Gelatin-Induced Angiogenic Gene Regulations

Further experiments compared the transcriptomes of ASC cultured in a gelatin sponge in the presence or absence of cilengitide. The addition of cilengitide induced significant regulations. The most significant ones (fold change > 10 and <10, *p* < 0.05) were functionally assigned by the Metascape gene annotation resource and included the following GO biological properties: vasculature development, circulatory system process, and response to wounding (Figure 7A). A total of 20 transcripts were found and among them, IL-8 (CXCL-8), IL-1, IL-6, CCL-2, and CCL-20 were downregulated, confirming that inflammatory/angiogenic factors were involved (Figure 7B). Q-RT PCR-amplifications confirmed that IL-1, IL-6, IL-8, and VEGF expression was reduced in ASC (cil)-gelatin sponge compared to control (Figure 7C). Thus, RGD motifs were implicated in the gelatin-induced angiogenic gene regulation of ASC.

### 2.5. In Gelatin, RGD-Binding Is Involved in the Secretion of Factors Regulating Endothelial Cell Migration, Tubulogenesis, and Permeability

The functional assays evaluating endothelial cell migration, tubulogenesis, and permeability upon ASC secretomes were performed after pre-blocking ASC with cilengitide. Importantly, ASCs were previously exposed to cilengitide and thereafter rinsed to exclude the presence of cilengitide in the conditioned media, RGD being involved in the migration and tubulogenesis of endothelial cells. Pre-blocking with cilengitide decreased the tubulogenic effect on endothelial cells promoted by ASC in gelatin (Figure 8A). The representative images of HUVEC tubes are shown in Figure 8B. Pre-blocking with cilengitide decreased the effect of ASC/gelatin on endothelial cell migration through a fibronectin gel (Figure 8C). Pictures showing crystal violet-labeled endothelial cells having crossed the gel are shown in Figure 8D. Regarding endothelial permeability, cilengitide decreased the ability of ASC to reduce the permeability of HUVEC to FITC-dextran (Figure 8E). Together, these observations indicate that RGD motifs are involved in the promotion of endothelial tubulogenesis and migration by ASC exposed to gelatin. Blocking RGD-binding integrins with cilengitide decreased the ability of ASC to reduce endothelial permeability and stabilize vessel structures.

### 2.6. In Gelatin, RGD-Binding Is Involved in the Secretion of Factors Promoting Vasculogenesis in Human Vascular Organoids

To further analyze endothelial cell functions, a more integrative model mimicking angiogenesis was employed using vascular organoids derived from human embryonic Pluripotent Stem Cells (ePSCs). The experiment followed a two-step differentiation protocol: the first step, lasting two weeks, focused on inducing the endothelial differentiation of the ePSC, while the second step, lasting one week, promoted vasculogenesis. Conditioned media from ASC/gelatin sponges versus ASC(cil)-gelatin sponges were applied during the second step to specifically study vasculogenesis rather than endothelial differentiation. A medium without angiogenic factors and medium supplemented with VEGF/PDGF were used as controls (Figure 9A). Fifty vascular organoids per condition, from separate experiments, were individually analyzed for both CD31-positive tubular and linear vascular structures. While some vascular structures were present in the two control media at day 21, vascular organoids exposed to a medium conditioned by ASC/gelatin sponges exhibited a higher number of vascular structures. The pre-treatment of ASC with cilengitide prevented this effect, confirming the involvement of RGD (Figure 9). Representative microscopic images are shown in Figure 9B (low magnification) and Figure 9C (higher magnification). Statistical analyses are presented in Figure 9D. Together, these observations confirmed that in a human vascular organoid model, secretions from ASC in gelatin sponges promoted vasculogenesis via RGD exposure.

## 3. Discussion

RGD motifs are central to cell-ECM interactions, mediating cell adhesion, migration, and survival through their interaction with integrins. Integrins such as α5β1 and αvβ3 play a pivotal role in angiogenesis by recognizing RGD sequences in ECM proteins like fibronectin and vitronectin [21]. These interactions activate intracellular signaling pathways that regulate angiogenesis and tissue repair, including the promotion of endothelial cell migration and tubulogenesis [22]. Current knowledge indicates that RGD-binding integrins facilitate the migration of mesenchymal and endothelial cells to sites of injury, where they promote neovascularization [9].

Our study builds on this understanding by showing that the exposure of ASC to RGD motifs in a gelatin matrix significantly enhances their angiogenic potential. The need for specificity controls explored the possible use of scaffolds that do not expose RGD motifs to ASC. Experiments (not shown) with collagen sponges (exposing few RGD, in a cryptic form compared to gelatin), decellularized matrix, ASC clusters in 3D with the same ASC–medium ratio, and ASC spheroids clearly indicated that each environmental condition created by the scaffolds or setups uniquely regulated ASC behavior. This was evidenced by differences in the microtopography of ASC within the scaffold structures. For instance, the Principal Component Analysis (Figure 1) confirmed significant differences in the transcriptomes of ASC spheroids, ASC in gelatin sponges, and ASC in 3D culture without scaffolds. Given these findings, we determined that using a scaffold without gelatin but with identical environmental parameters (shape, volume, porosity, pore size, etc.) was not scientifically feasible. Instead, the most appropriate specificity control for this study was the use of ASC cultured in gelatin sponges with RGD-binding integrins pre-blocked by Cilengitide.

The upregulation of angiogenic factors genes, including *CXCL3*, *adrenomedullin*, *HGF*, *IL-6*, *ANGPT1*, *CXCL1*, *CXCL5*, *IL-1β*, and *IL-8*, in ASC in response to gelatin exposure highlights the dynamic role of these cells in modulating vascularization in tissue repair and regeneration. CXCL3, CXCL1, and CXCL5 are critical chemokines that recruit neutrophils and endothelial progenitor cells to sites of injury, facilitating an early angiogenic response. Adrenomedullin (ADM), a vasodilatory peptide, enhances endothelial survival and tube formation, contributing to vascular homeostasis under inflammatory conditions. HGF, a well-characterized angiogenic growth factor, promotes endothelial cell proliferation and migration, underscoring its relevance in tissue repair. IL-6 and IL-1β, while traditionally associated with inflammation, also exhibit pro-angiogenic functions by modulating endothelial cell activation and enhancing vascular permeability. IL-8 serves as a potent chemoattractant for neutrophils and endothelial cells, amplifying the angiogenic cascade. ANGPT1, through its interaction with the Tie2 receptor, stabilizes newly formed vessels, fostering the maturation of functional vascular networks. Together, these factors form a coordinated network that supports the initiation, progression, and stabilization of angiogenesis. Physiologically, such upregulation suggests that gelatin may act as a biocompatible scaffold that primes ASC to secrete a robust angiogenic secretome. This response may have evolved as a mechanism to ensure efficient vascularization during tissue repair, particularly in wounds requiring extracellular matrix remodeling. This aligns with the role of ASC in promoting vascular remodeling and healing, offering potential translational implications for regenerative therapies targeting ischemic or poorly vascularized tissues. The increase in endothelial cell migration and tube formation observed in vitro supports the idea that ASC’s interaction with RGD-rich ECM (typically a healing situation) components enhances their ability to facilitate angiogenesis. Previous studies have highlighted the importance of fibronectin-RGD interactions in promoting angiogenesis, but our findings show that ASCs also respond to RGD exposure by adopting a more angiogenic phenotype, likely enhancing their effectiveness in tissue repair settings.

The dual roles of inflammation and angiogenesis in tissue repair have been well documented, but the interplay between these processes, particularly in the context of ASC function, remains complex. Inflammation is generally considered a precursor to angiogenesis, with inflammatory cytokines such as IL-1, IL-6, and IL-8 playing important roles in initiating angiogenesis by recruiting endothelial progenitor cells and promoting their migration. At the same time, excessive inflammation can impair tissue regeneration, necessitating a fine balance between pro-inflammatory signals and pro-angiogenic responses. Our study suggests that ASC exposed to RGD motifs can balance these two processes. We observed that ASC upregulated several inflammatory cytokines, including IL-1 and IL-6, while also increasing the expression of VEGF and other pro-angiogenic factors. This suggests that ASCs exposed to RGD are primed to support both the inflammatory and angiogenic phases of wound healing. Further investigation could provide insights into how ASCs fine-tune their response to the wound microenvironment.

Which component of angiogenesis is regulated? Angiogenesis is a complex, multi-step process that includes endothelial cell proliferation, migration, differentiation into tubular structures, and vessel stabilization [23]. Our findings show that RGD exposure primarily enhances endothelial cell migration and tubulogenesis while having no significant effect on proliferation. This suggests that the RGD-integrin interaction is more closely associated with the later stages of angiogenesis, where endothelial cells migrate toward angiogenic stimuli and organize into tubular networks. Previous research has demonstrated that integrins, particularly αvβ3 and α5β1, are key regulators of cell migration and tubulogenesis, both of which are critical for neovascularization during tissue repair [24]. The lack of a proliferative response in endothelial cells exposed to ASC-conditioned media suggests that ASC may not directly promote the early stages of angiogenesis, which involve rapid cell division. Instead, ASCs seem to play a more prominent role in guiding endothelial cell migration and supporting vessel maturation, as evidenced by their influence on endothelial permeability. Vessel stabilization, which involves the formation of tight junctions and decreased endothelial permeability, is crucial for maintaining the integrity of new blood vessels. The ability of ASC to reduce endothelial permeability, as observed in our study, suggests that they also contribute to the final stabilization and maturation of new vessels [25]. The differential regulation of endothelial functions by ASC exposed to RGD motifs highlights the specific roles of RGD-binding integrins in mediating various aspects of angiogenesis. While ASC-conditioned media did not affect endothelial proliferation, it significantly enhanced migration and tubulogenesis, key functions regulated by integrin signaling. This is consistent with the known role of integrins in modulating cell migration through focal adhesion kinase (FAK)-mediated signaling pathways, which are activated upon integrin engagement with RGD motifs [22]. FAK activation leads to cytoskeletal rearrangements and increased motility, allowing endothelial cells to migrate toward sites of angiogenesis [26]. The promotion of endothelial tubulogenesis by ASC further underscores the importance of RGD-binding integrins in regulating endothelial cell differentiation into capillary structures. Studies have shown that RGD-containing ECM proteins, such as fibronectin and vitronectin, promote tubulogenesis by activating integrin-dependent signaling pathways that regulate the cytoskeletal organization and cell–cell interactions [27]. Our findings suggest that ASC exposed to RGD motifs enhance these processes, likely by secreting factors that promote endothelial cell organization and stabilize newly formed vessels. Interestingly, the study also found that ASC reduced endothelial permeability, which is essential for the stabilization of blood vessels. Endothelial permeability is dynamically regulated during angiogenesis, with increased permeability allowing for the infiltration of immune cells during inflammation, and decreased permeability contributing to vessel maturation and stability [28]. The ability of ASC to decrease endothelial permeability suggests that they play a role in the later stages of angiogenesis, where vessel stability is crucial for functional tissue regeneration. Together, these findings suggest multiple facets of the angiogenic process activated by RGD-stimulated ASC. Supporting this, previous in vivo studies have demonstrated that rat ASC in a gelatin sponge promotes neovascularization in a model of ischemic skin defects [18].

Which RGD-binding integrins could be involved? To investigate the specific roles of the three main RGD-binding integrins—α5β1, αvβ3, and αvβ5—we also tested additional inhibitors alongside cilengitide: volociximab (anti-α5β1), intetumumab (anti-αv), and etaracizumab (anti-αvβ3) (not shown). We observed that volociximab and etaracizumab did not prevent ASC attachment to gelatin, and that intetumumab failed to reverse some of the endothelial functions studied. These findings indicate that the roles of RGD-binding integrins are highly complex, with probably significant compensatory effects between receptors. Additionally, the specificity of these inhibitors remains incompletely understood, further complicating data interpretation. For this reason, further studies would be necessary to precisely define the role of each integrin.

The vascular organoid model derived from ePSC provides a more physiologically relevant system for mimicking angiogenesis compared to traditional endothelial cell-only models. It offers distinct advantages for studying angiogenesis in a fully human context, avoiding species-specific differences inherent in animal models. Additionally, it provides high reproducibility, making it a valuable tool for modeling angiogenesis [29]. However, this model lacks systemic interactions, such as those involving hormonal or hemodynamic influences, which are crucial for understanding angiogenesis in vivo. Furthermore, it does not replicate the complexity of immune responses, true vascular perfusion, or chronic conditions, all of which are better studied in animal models. Alternative in vitro models of angiogenesis in further studies could combine organoid-based assays with microfluidics, stromal cells, and immune cells such as “vascular-on-a-chip” to recreate dynamic flow conditions, enabling the study of angiogenesis under shear stress, nutrient gradients, or immune cells infiltration. Our additional results, which demonstrate that ASC exposed to gelatin sponges significantly enhanced vasculogenesis, underline the advantages of using this 3D organoid system. In contrast to the study of endothelial functions, which are limited in capturing the full complexity of angiogenesis, vascular organoids create an environment that closely mimics in vivo tissue, incorporating multiple cell types and ECM components. This allows for the observation of both te early and late stages of blood vessel formation, including the organization of vascular structures and interactions with ECM. ASC conditioned by gelatin sponges were shown to promote the CD31+ tubular/linear structures in favor of vasculogenesis. The involvement of RGD motifs, confirmed by cilengitide pre-treatment, highlights how specific integrin-mediated interactions contribute to vasculogenesis. By replicating the 3D structure of tissues, this model enabled the observation of not only increased endothelial cell migration, tubulogenesis, and permeability, but also the maturation and stabilization of newly formed vascular structures.

We used a gelatin sponge to mimic the provisional matrix during healing. While gelatin sponges offer a useful possibility to mimic the provisional matrix observed in wound healing, characterized by denatured collagen fibers, they have notable limitations. One key drawback is the lack of granulation tissue infiltration, which is critical in vivo for the wound-healing process. Granulation tissue is rich in immune, inflammatory, and endothelial cells that coordinate extracellular matrix remodeling, angiogenesis, and immune responses. The absence of these cell populations in gelatin sponge models limits their ability to fully replicate the dynamic cellular interactions and signaling cascades occurring in a healing wound, thereby constraining their translational relevance.

Addressing scalability is critical for the clinical application of ASC/gelatin-based therapies. Strategies such as advanced bioreactor systems are pivotal for expanding ASC while preserving their introduction within gelatin scaffolds. This scalability is expected to enable large-scale production under Good Manufacturing Practice conditions, meeting regulatory standards for clinical use. Additionally, the use of a gelatin sponge as a matrix offers advantages for clinical translation. Gelatin sponge is indeed biocompatible, biodegradable, and already widely used in medical applications, such as hemostatic agents and drug delivery systems. Its RGD-exposing properties not only support ASC adhesion and activation but also facilitate integration into wound sites and infiltration by host inflammatory/angiogenic cells. While gelatin sponges offer significant advantages as RGD-delivering scaffolds, alternative materials could also be explored for therapeutic applications such as synthetic or natural matrices (like fibrin and fibronectin) to provide tunable mechanical properties, controlled degradation rates, and tailored RGD presentation. These scaffolds may be better suited for specific tissue environments or applications requiring prolonged support. These attributes make gelatin sponges a promising delivery vehicle for ASC in diverse therapeutic applications, including tissue repair and vascular regeneration.

## 4. Conclusions

In vivo, ASCs are likely to encounter RGD motifs in the provisional ECM during the early stages of wound healing. This matrix, which is rich in fibronectin and vitronectin, and MMPs exposing RGD motifs in collagen fibers through degradation, provides the necessary cues for ASC recruitment and activation. Our findings suggest that exposure to RGD primes ASCs for enhanced pro-angiogenic activity, enabling them to contribute to the neovascularization process more effectively. By promoting endothelial cell migration and tubulogenesis, ASCs can support the formation of new blood vessels, which are essential for supplying oxygen and nutrients to the regenerating tissue. Furthermore, the ability of ASCs to modulate endothelial permeability in vivo may help ensure that newly formed vessels are stabilized and integrated into the existing vasculature. This suggests that ASCs are not only involved in the initiation of angiogenesis but also play a critical role in the resolution and maturation phases of vessel formation. These functions are likely to be crucial for the successful integration of ASCs into therapeutic applications aimed at promoting tissue repair and regeneration, particularly in ischemic or damaged tissues where rapid revascularization is needed [30].

## 5. Materials and Methods

Reagents: Cilengitide trifluoroacetic acid salt was purchased from Sigma-Aldrich (Saint Louis, MO, USA), porcine gelatin powder from Sigma-Aldrich (Saint Louis, MO, USA).

Preparation of gelatin gels: Gelatin powder from porcine skin (Sigma Aldrich (Saint Louis, MO, USA)) was weighed and dissolved in PBS at 50 °C and sterilized as rapidly as possible through 0.22 µm filters. In ASC/gelatin gel binding experiments, gelatin solution was added to microbial transglutaminase (Sigma Aldrich, Saint Louis, MO, USA). In this case, this solution was heated not higher than 70 °C. Ten units of transglutaminase per gram of gelatin was used. Cross-linked gels were incubated at 37 °C for 4 and heat-treated in PBS for 30 min at 65 °C to inactivate the remaining enzyme.

ASC lines: All the ASCs were derived from the subcutaneous fat of donors. The ASC lines used in this study were fully validated for their phenotype (CD14− CD44+ CD45− CD73+ CD90+ CD105+ HLA-DR−), multipotency (osteocytic, adipogenic, and chondrocytic differentiation), and regenerative potential and derived upon the ethical authorization 2020-01102 and NAC 14-183. For the transcriptomics and proteomics experiments, the following lines/donors were used: #1 (female, 50 years old); #2 (male, 64 years old), and #3 (male, 49 years old). For the assays using HUVEC, line#5 was used (female, 63 years old).

ASC culture: ASCs were prepared between passages 2–5 and were cultured in Dulbecco’s Modified Eagle Medium (DMEM) with 4.5 g/L glucose and L-Glutamine, supplemented with 10% human platelet lysate (MultiPL100—Macopharma, Tourcoing, France) and 1% penicillin–streptomycin (ThermoFisher, Waltham, MA, USA) at 37 °C and under 5% CO_2_. To manufacture the ASC/gelatin sponge, a piece of sterile absorbable gelatin sponge USP Spongostan (standard, Ethicon, Raritan, NJ, USA) of 840 mm^3^ was soaked in a suspension (300 µL) of ASCs at a final density of 6000 cells/mm^3^ in the ASC culture medium (optimal density for saturation). The ASC/gelatin sponge was then cultured from 24 h to 8 days in air/liquid interface conditions using a Millipore insert (polytetrafluoroethylene, 30 mm–0.4 µm pore, Millipore corporation, Burlington, MA, USA) floating on 1 mL of ASC culture medium in a 6-well plate. Medium change was performed every 2 days. In the transcriptomics experiments, the gelatin sponge was removed, with the same volume and number of cells being deposited directly on the membrane to form a compact tissue. In experiments with RGD inhibition (study of the ASC secretome’s impact on HUVEC functions or vascularized organoids), ASCs were pre-incubated with cilengitide 7.5 mg/mL for 1 h at room temperature (optimal incubation time) and rinsed twice in culture medium prior to their integration within the matrix.

Molecular biology: A microarray was used as the best way to simply analyze the global cell regulation within a gelatin sponge environment. The isolation of total RNA was performed by using an RNeasy kit from Qiagen (Hombrechtikon, Switzerland) according to the manufacturer’s instructions. RNA concentration was determined by a spectrometer (Thermo Scientific™ NanoDrop 2000, ThermoFisher, Waltham, MA, USA), and RNA quality was verified by a 2100 bioanalyzer (Agilent, Santa Clara, CA, USA). Human microarray was performed with the ClariomTM S Assay for humans (ThermoFisher, Waltham, MA, USA) using the Complete GeneChip^®^ Instrument System, Affymetrix. The Principal Component Analysis was computed using the TAC4.0.1.36 software (Biosystems, Muttenz, Switzerland) with default settings. The Gene Set Enrichment Analysis (GSEA) was used to analyze the pattern of differential gene expression between the human ASC patch and the monolayer condition. The Gene Ontology Biological Process (GOBP) gene set from the Molecular Signatures Database was used. The enrichment of processes and pathways within the significantly regulated transcripts (fold change > 2, FDR < 0.01) identified between various conditions was assessed using Metascape (www.metascape.org, accessed on 11 March 2022).

Mass spectrometry: Serum-free media (ASC culture medium without human platelet lysate) were conditioned by ASC suspensions or ASC in gelatin or gelatin without ASC. Upon the clarification of supernatants at 500× *g* for 10 min, proteins were precipitated and digested, and peptides were analyzed by nanoLC-MSMS using an easynLC1000 (ThermoFisher, Waltham, MA, USA) coupled with a Q Exactive HF mass spectrometer (ThermoFisher, Waltham, MA, USA). Database searches were performed with Mascot (Matrix Science, London, UK) using the Human Reference Proteome database (Uniprot). Data were analyzed and validated with Scaffold (Proteome Software, Portland, OR, USA) with 1% of protein FDR and at least 2 unique peptides per protein with a 0.1% of peptide FDR.

Migration, tubulogenesis, and permeability of endothelial cells: The migration, tubulogenesis, and permeability of Human Umbilical Veinous Endothelial Cells (HUVECs) were explored to determine the functional impact of the ASC/gelatin secretome on endothelial cells. The HUVECs were cultivated in complete endothelial cell medium 2 (both from Sigma, Buchs, Switzerland). The migration of the HUVECs was analyzed using the endothelial cells migration assay (Sigma, Buchs, Switzerland) following the manufacturer’s instructions. Briefly, the HUVECs were starved for 15 h in endothelial cell medium 2 without serum and supplements and then introduced into a Boyden chamber with a semi-permeable membrane coated with fibronectin, or bovine serum albumin (BSA) as a control, at the bottom. Migration towards supplement-free endothelial cell medium 2, conditioned for 48 h by ASC, was measured by staining the cells with crystal violet and extracting the dye that migrated outside the Boyden chamber (via the measurement of the absorbance of the extract at 540 nm). Migration was quantified as the difference between the absorbance with fibronectin and the absorbance with control BSA. For the tubulogenesis assay, serum/supplement-free endothelial cell medium 2 was conditioned for 48 h with ASC. The analysis of the tubular assembly of the HUVECs was conducted using the angiogenesis assay kit (Abcam, Cambridge, UK) in accordance with the manufacturer’s instructions. Briefly, the HUVECs were plated in their conditioned medium on a gel containing fibronectin for 24 h, followed by cell staining with a fluorescent dye and tube analysis via the Cytation 5 cell imaging reader (Agilent, Santa Clara, CA, USA). The permeability of endothelial cells was assessed using the in vitro vascular permeability assay (Millipore) according to the manufacturer’s instructions. Briefly, a monolayer of the HUVECs was established in 96-well plate inserts (Boyden chamber) before the addition of media conditioned by ASC in a gelatin sponge. Under some conditions, ASCs were pre-incubated with inhibitors for 1 h at room temperature and washed twice in an ASC culture medium. The permeability of the HUVEC layer was measured by its ability to allow FITC-dextran diffusion outside the Boyden chamber.

Generation of vascular organoids from embryonic pluripotent stem cells: The human embryonic stem (hESCs) cell line HS420 (Gift from Dr Outi Hovatta, Karolinska institute, Sweden) was cultured in Stemflex medium (Thermofisher) on laminin 521-coated tissue culture flasks (Thermofisher) according to manufacturer’s instructions. The HS420 cells at 70% of confluency were enzymatically passaged using Accutase (Thermofisher, ref 00-4555-56) in Aggrewell TM (Stemcell technologies, Basel, Switzerland) culture plates in Stemflex medium (Thermofisher) supplemented with 1% of PenStep and ROCK inhibitor (Y27632, abcam) at 10 μM. In parallel, ASCs were resuspended in 20 mL of basal vascular medium (IMDM/DMDM F12) for further processing. For the experimental conditions, the cells were prepared to achieve a concentration of 5 × 10^6^ cells/mL. In the gelatin sponge condition, ASCs were added to a gelatin sponge, while the cilengitide condition involved the pre-treatment and rinsing of ASCs by cilengitide, which required incubation with gentle agitation. After one hour of incubation, the cilengitide-treated cells were rinsed with vascular medium, centrifuged, and resuspended in 1.5 mL of the same medium for loading into sponges. The sponges were then loaded with 300 µL of the cell suspension, and 2 mL of vascular medium was added under the inserts to create optimal conditions. Finally, after 96 h, the medium was recovered from the inserts for further use in the culture of vascular organoids. Next, the HS420 cells were deposited in supplemented StemFlex medium within Aggrewell-800™ plates (Stem Cell Technologies, Basel, Switzerland) at a density of 2000 cells per microwell. To ensure the proper distribution of the cells, the plate was gently shaken and placed on a stable support. After 15 min, the plate was cultured at 37 °C for 24 h to facilitate organoid formation. Following this period, the resulting spheres were collected and transferred to a standard 6-well plate containing Dulbecco’s Modified Eagle Medium/Nutrient Mixture F-12 (DMEM-F12, ThermoFisher, Waltham, MA, USA) supplemented with 1% PenStep, 1% B-27 supplement (ThermoFisher, Waltham, MA, USA), 1% N2 supplement (ThermoFisher, Waltham, MA, USA), 8 µM CHIR99021 (Axon Medchem, Reston, VA, USA), and 25 ng/mL BMP4 (ThermoFisher, Waltham, MA, USA). The organoids were then cultured with constant agitation in 3 to 4 mL of medium per well at 37 °C and 5% CO_2_, using an orbital shaker set at 60 rpm. On day 3, the medium was replaced with a combination of half DMEM-F12 and half Iscove’s Modified Dulbecco’s Medium (IMDM, ThermoFisher, Waltham, MA, USA), supplemented with 25 ng/mL VEGF (ThermoFisher, Waltham, MA, USA), 10 ng/mL PDGF (Cell Signaling, Danvers, MA, USA), and 2 ng/mL Activin A (Cell Signaling, Danvers, MA, USA). From days 7 to 14, the organoids continued to be cultivated in a mixture of half DMEM-F12 and half IMDM, supplemented with 25 ng/mL VEGF and 10 ng/mL PDGF. From days 14 to 21, vascular organoids in the positive control group were maintained in a basal vascular medium composed of half DMEM-F12 and half IMDM, supplemented with 25 ng/mL VEGF and 10 ng/mL PDGF. In contrast, the negative control group had their vascular organoids maintained in the basal vascular medium without any supplements. For the ASC/gelatin sponge groups, the organoids were cultivated in a basal medium that had been conditioned for 96 h, derived solely from gelatin sponge loaded with ASCs or gelatin sponge loaded with ASCs pre-treated by cilengitide. The culture medium was changed every 2–3 days over the 21-day culture period to ensure optimal growth conditions. Morphologically, the organoids should display a clear, defined external layer, which further reflects healthy differentiation and immunostaining for CD31, SMA, and SM22 ensuring vascular differentiation.

Immunohistochemistry: Vascular organoids were fixed in 4% paraformaldehyde at room temperature for 45 min and subsequently dehydrated using a series of increasing alcohol concentrations, followed by xylene. The organoids were embedded in paraffin blocks and sectioned into 3 μm slices. The sections were mounted on glass slides, deparaffinized with xylene, and rehydrated through a series of decreasing alcohol concentrations. To unmask antigens, the slides were boiled in 0.01 M sodium citrate buffer (pH 6.0) for 15 min. The samples were then washed in 1× PBS for 5 min, followed by permeabilization through incubation with 0.2% PBS/Triton for 15 min. Subsequently, the sections were incubated for 10 min in 2.5% Normal Horse Serum to block non-specific binding. The sections were then incubated overnight with a primary antibody, diluted to 1:4000 for anti-CD31 (ab281583, Abcam) in a PBS buffer containing 1.5% blocking serum. After washing in PBS, the sections were incubated for 10 min with a Biotinylated Pan-Specific Universal Antibody (PK-7800, Vector Laboratories, Newark, CA, USA), followed by another wash in PBS. The sections were then treated with a Streptavidin/peroxidase complex for 5 min and washed again in PBS. They were subsequently incubated in a peroxidase substrate solution (SK-4105, Vector Laboratories) until the desired staining intensity developed. After rinsing in tap water, the slides were stained for 5 min in a Hematoxylin solution, followed by rinsing in running tap water and deionized water. The slides were then dehydrated in alcohol, cleared in xylene, and mounted. After the staining procedure, the slides were imaged using an Axioscan microscope (Zeiss Axioscan.Z1, Zeiss, Oberkochen, Germany). The obtained images were then analyzed manually in a blinded manner to detect all the vascular structures using the QuPath 0.5.1 software.

Statistical analysis: Statistical analysis was performed using GraphPad Prism version 6.0 (GraphPad Software, La Jolla, CA, USA). *p* values less than 0.05 were considered statistically significant, and were indicated as follows: *: *p* < 0.05; **: *p* < 0.01; ***: *p* < 0.001 (non-parametric Mann–Whitney test, suitable for comparing independent groups without a normal distribution).

Experimental groups: The transcriptomics and proteomics experiments were performed using 3 independent ASC lines. Studies evaluating the impact of conditioned media on HUVEC tubulogenesis: n = 8, 2 independent experiments; studies evaluating the impact of conditioned media on HUVEC migration: n = 4, 2 independent experiments; studies evaluating the impact of conditioned media on HUVEC permeability: n = 6, 3 independent experiments; studies evaluating the impact of conditioned media on vascularized organoids: n = 50, 2 independent experiments.

## Figures and Tables

**Figure 1 ijms-26-00867-f001:**
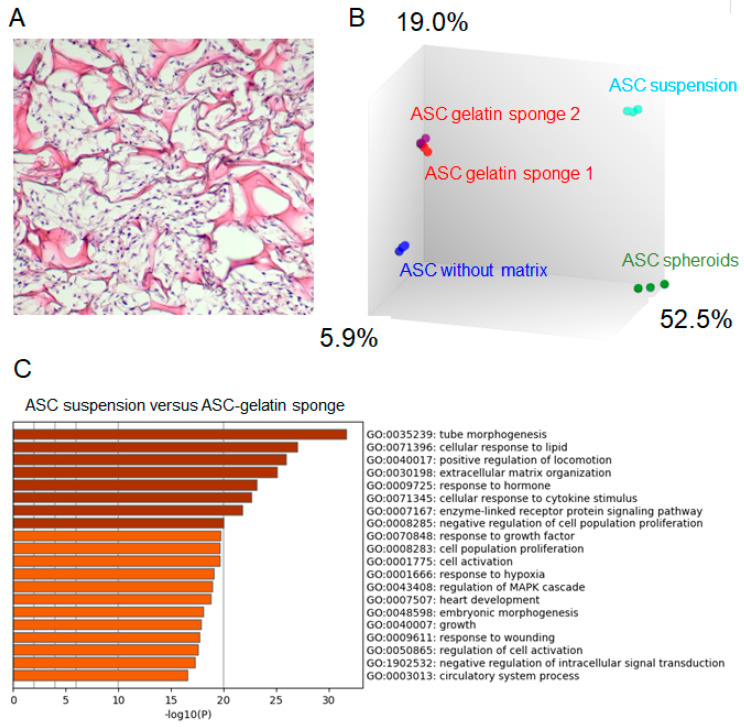
ASC cultured in a porcine gelatin sponge shows angiogenic gene regulation. Three ASC lines from three independent donors were seeded into a porcine gelatin sponge, clustered into 3D spheroids by forced aggregation, or seeded onto a semi-permeable membrane to form a 3D sheet of cells. All of these tissues were cultured in air/liquid interface conditions. A single-cell suspension of ASC from a monolayer culture was used as a comparator. (**A**) The Hematoxylin and eosin staining of a tissue section from ASC culture in a gelatin sponge. (**B**) The Principal Component Analysis of ASC transcriptomes under all the conditions. Three independent ASC lines per condition are shown. The indicated percentages refer to the proportion of the total variance in the dataset that is explained by each principal component (**C**) According to the gene analysis and annotation resource Metascape (metascape.org), functional families associated with the most regulated genes between ASC/gelatin sponge and ASC monolayer (fold change > 5 and <5, *p* < 0.05) are identified. The enrichment analysis of statistically enriched GO Biological Process (BP) terms is shown. The parameters used for the analysis were as follows: Organisms: Homo Sapiens, Input gene set: GO Biological Process; Min Overlap: 3; *p* value cutoff: 0.01; Min enrichment: 0.01.

**Figure 2 ijms-26-00867-f002:**
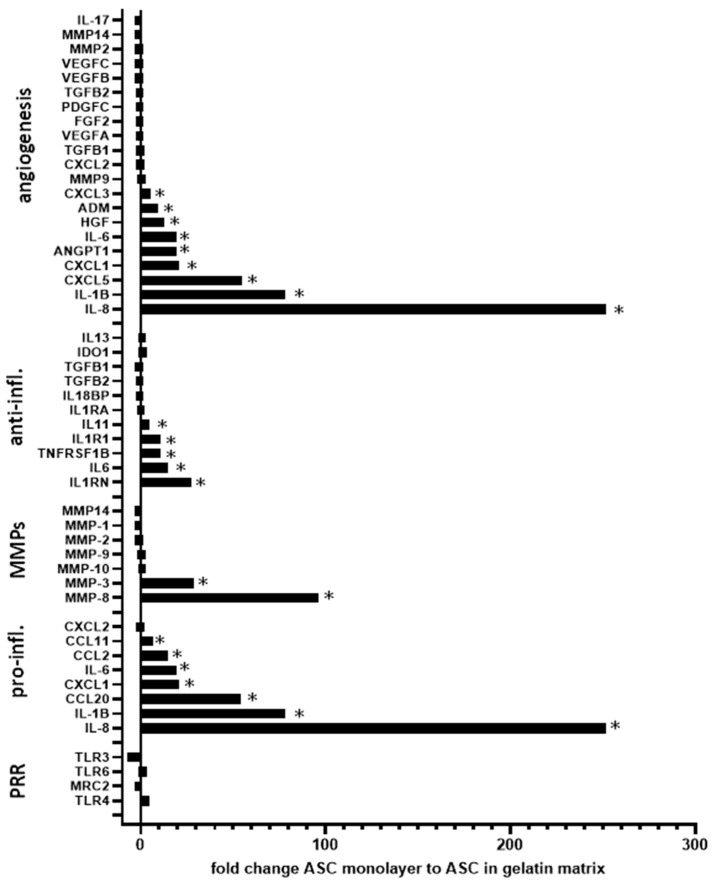
Genes linked to inflammation, angiogenesis, and healing are upregulated when ASC are cultured in a gelatin sponge compared to ASC suspension. Three ASC lines from three independent donors were seeded into a porcine gelatin sponge or prepared as single-cell suspensions from a monolayer culture. A fold change analysis for a selected list of factors related to healing, inflammation, and angiogenesis was performed using the Clariom™ S Assay for humans (ThermoFisher, Waltham, MA, USA) with the Complete GeneChip^®^ Instrument System, Affymetrix. The fold change analysis was computed using the TAC 4.0.1.36 software (Biosystems, Muttenz, Switzerland) with default settings. * corrected *p* value < 0.05.

**Figure 3 ijms-26-00867-f003:**
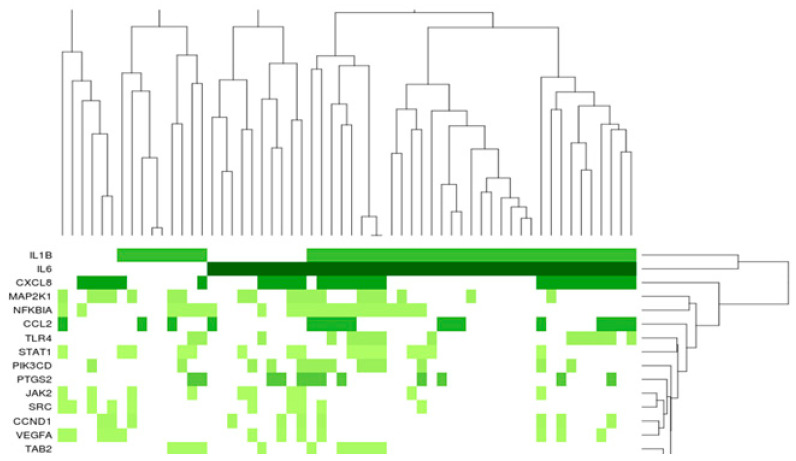
Gene Set Enrichment Analysis (GSEA) confirmed the importance of inflammatory/angiogenic factors in the ASC regulation induced by culture in a gelatin sponge. Three ASC lines from three independent donors were seeded into a porcine gelatin sponge or prepared as single-cell suspensions from a monolayer culture. Full gene expression profiles for each condition were established using the Clariom™ S Assay for humans (ThermoFisher, Waltham, MA, USA) with the Complete GeneChip^®^ Instrument System, Affymetrix. GSEA was performed using Cytoscape 3.8.2 (Moutain view, CA, USA). Shades of green reflect gene expression levels (log-fold changes). Lighter green represents lower log-fold change.

**Figure 4 ijms-26-00867-f004:**
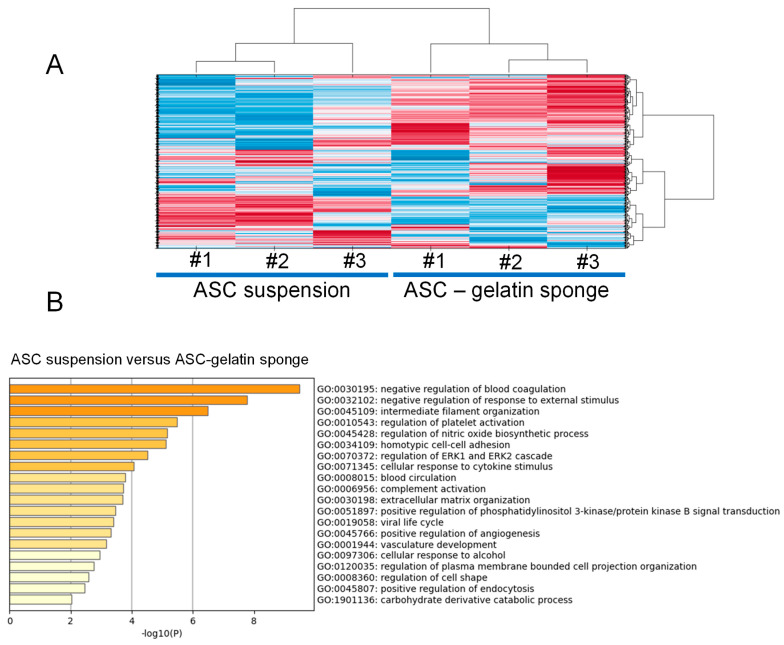
Mass spectrometry shows that ASC cultured in a gelatin sponge modify their secreted proteome in favor of angiogenesis. Three independent ASC lines were cultured in monolayer or within a porcine gelatin sponge, for 24 h, in a protein-free medium. Upon the clarification of supernatants, the proteins were analyzed by nanoLC-MSMS coupled with a Q Exactive HF mass spectrometer. Database searches were performed with Mascot (Matrix Science, London, UK) using the Human Reference Proteome database (Uniprot). (**A**) Data were analyzed and validated with Scaffold (Proteome Software, Portland, OR, USA) with 1% protein FDR and at least 2 unique peptides per protein with 0.1% peptide FDR to establish a hierarchical clustering of the ASC lines between the 2 conditions. # indicates the number of each ASC line. (**B**). According to the gene analysis and annotation resource Metascape (metascape.org), functional families associated with the most regulated genes between ASC/gelatin sponge and ASC monolayer (fold change > 2 and <2, *p* < 0.05) are identified. The enrichment analysis of statistically enriched GO Biological Process (BP) terms is shown. The parameters used for the analysis were as follows: Organisms: Homo Sapiens, Input gene set: GO Biological Process; Min Overlap: 3; *p* value cutoff: 0.01; Min enrichment: 0.01.

**Figure 5 ijms-26-00867-f005:**
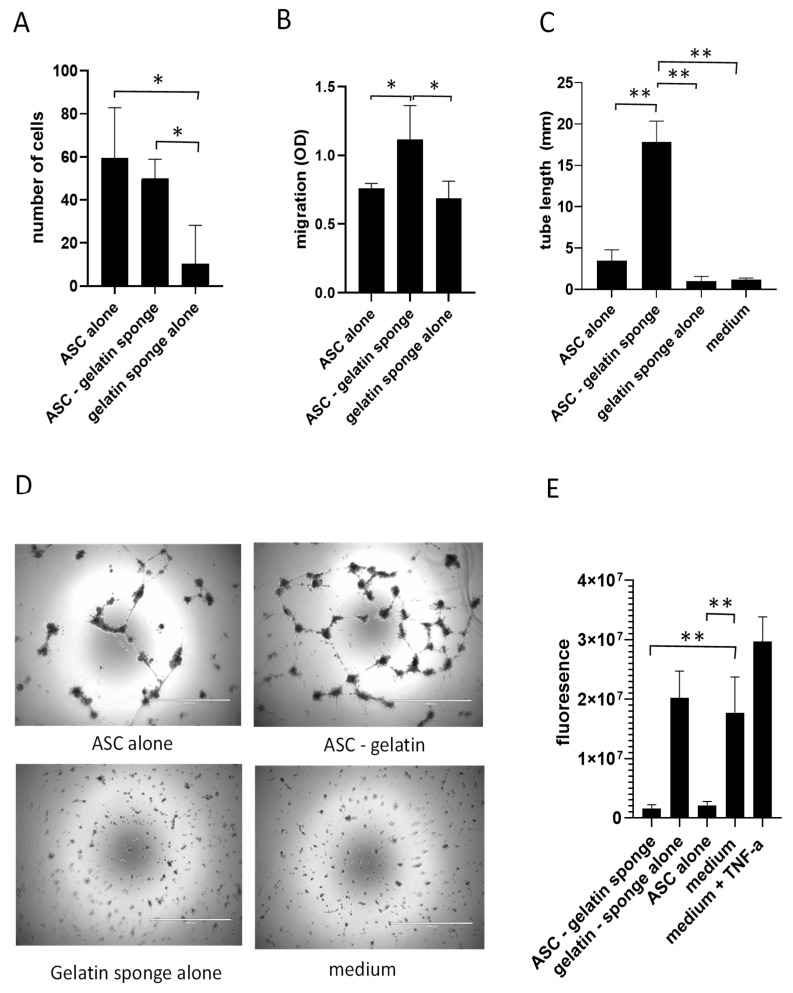
Media conditioned with ASC cultured in a gelatin sponge modify endothelial cell functions. The media conditioned with ASC monolayer culture, ASC/gelatin sponge, or gelatin sponge alone were exposed to HUVEC prior to the analysis of their proliferation (**A**), migration (**B**), tubular assembly (**C**), and permeability (**D**). (**A**) Proliferation was assessed after 48 h by cell counting using trypan blue exclusion. (**B**) Conditioned media were introduced outside a Bsoyden chamber while HUVEC were plated inside the Boyden chamber. Migration outside the Boyden chamber through a fibronectin layer was measured by colorimetry (crystal violet). Migration through a layer of bovine serum albumin was used as a controlled background and removed from the signal. (**C**,**D**) The conditioned medium was exposed to HUVEC cultured on an extracellular matrix gel for tubular assembly. A minimal medium without angiogenic factors was used to prevent spontaneous assembly. The total length of the formed tubes was quantified by an image analysis software (image J, version 1.54f). Representative pictures are shown in (**D**). (**E**) The conditioned media were exposed to a paucilayer of HUVEC with the aim of measuring endothelial permeability to a fluorescent probe (FITC-dextran). * *p* < 0.05; ** *p* < 0.001; Mann–Whitney.

**Figure 6 ijms-26-00867-f006:**
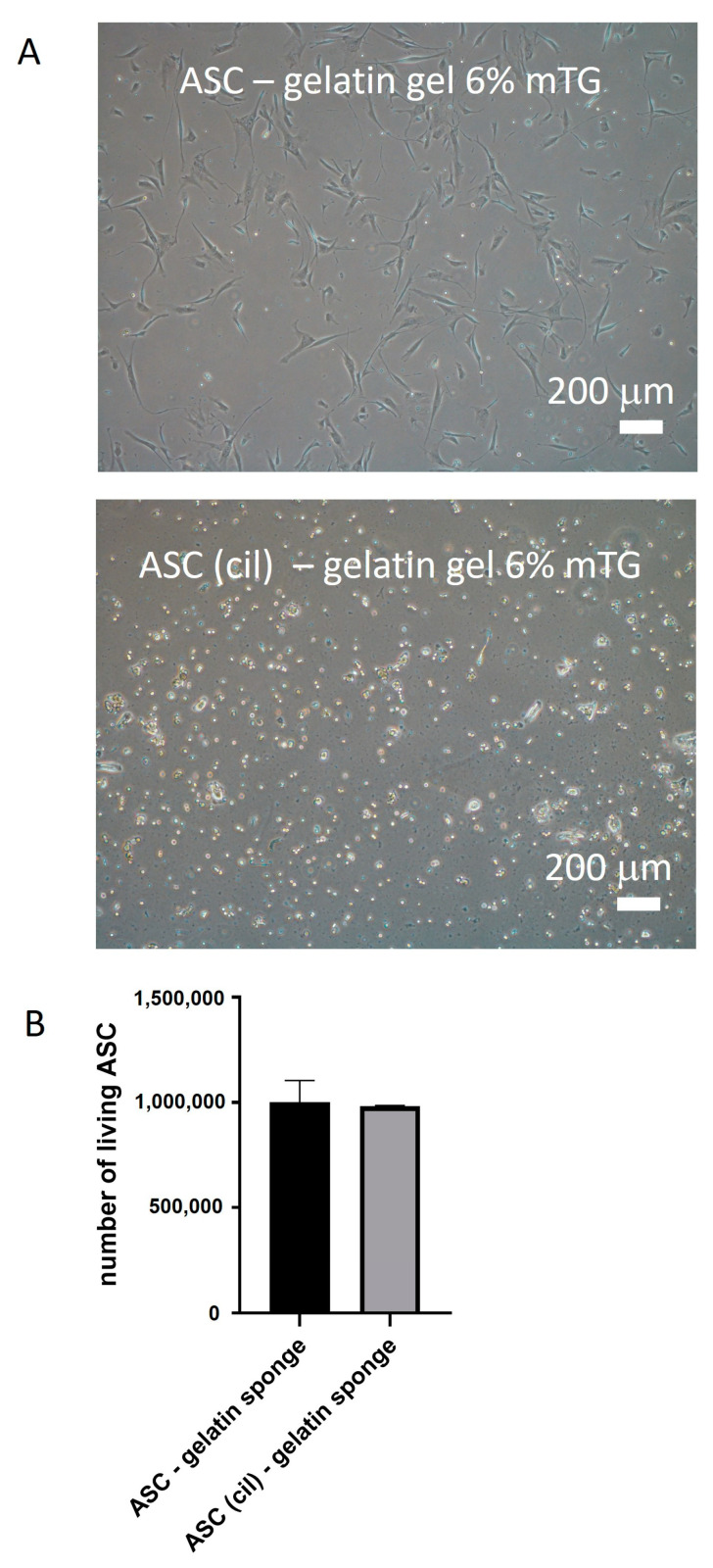
RGD motifs participate in ASC attachment to gelatin. A gelatin gel (6%) cross-linked with microbial transglutaminase was used for plating ASC in monolayer culture in the presence or not of cilengitide. A microscopic view of cell culture is shown in (**A**). (**B**) The cell quantification of ASC exposed to cilengitide, after 24 h, by using trypan blue exclusion.

**Figure 7 ijms-26-00867-f007:**
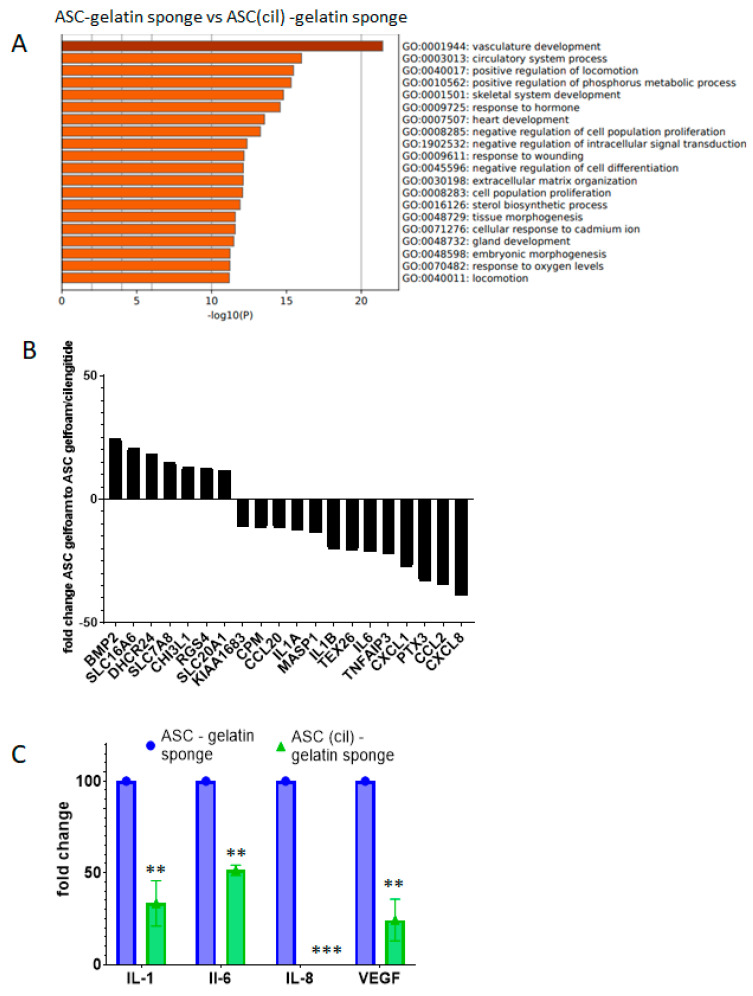
Exposure of ASC to cilengitide in a gelatin sponge induced gene regulations decreasing angiogenic/inflammatory factors. The full gene expression profiles of ASC/gelatin sponge versus ASC(cil)-gelatin sponge were established and compared. Biological triplicates were used, using three independent ASC lines. (**A**) A fold change analysis was calculated by the Complete GeneChip^®^ instrumentation system, Affymetrix, and Clariom^®^TAC4.0.1.36 software (Biosystems). According to the gene analysis and annotation resource Metascape (metascape.org), functional families associated with the most regulated genes between ASC/gelatin sponge and ASC monolayer (fold change > 10 and <10, *p* < 0.05) are identified. The enrichment analysis of the statistically enriched GO Biological Process (BP) terms is shown. The parameters used for the analysis were as follows: Organisms: Homo Sapiens, Input gene set: GO Biological Process; Min Overlap: 3; *p* value cutoff: 0.01; Min enrichment: 0.01. (**B**) List of the identified most regulated genes between ASC/gelatin sponge and ASC monolayer, with a statistical significance (*p* < 0.05) (fold change > 10 and <10), attributed to the calculated fold change. (**C**) The Q-RT PCR analysis of IL-1, IL-6, IL-8, and VEGF transcripts comparing ASC/gelatin sponge and ASC (cil)-gelatin sponge. ** *p* < 0.001; *** *p* < 0.0005; Mann–Whitney.

**Figure 8 ijms-26-00867-f008:**
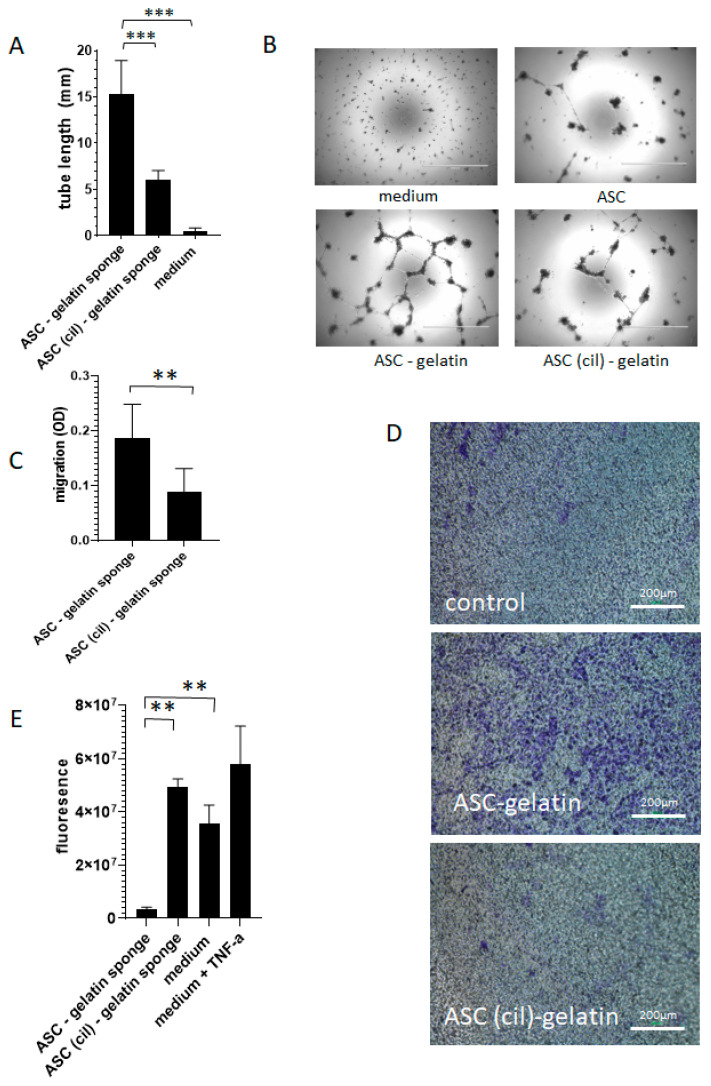
RGD motifs are involved in the secretion of factors from ASC in gelatin sponge influencing HUVEC functions. Media conditioned with ASC/gelatin sponge or ASC (cil)-gelatin sponge or gelatin sponge alone were exposed to HUVEC prior to the analysis of their tubular assembly (**A**,**B**), migration (**C**,**D**), and permeability (**E**). In (**E**), a medium with TNF-α 500 ng/mL was used as a positive control for increased permeability. ** *p* < 0.001; *** *p* < 0.0005; Mann–Whitney.

**Figure 9 ijms-26-00867-f009:**
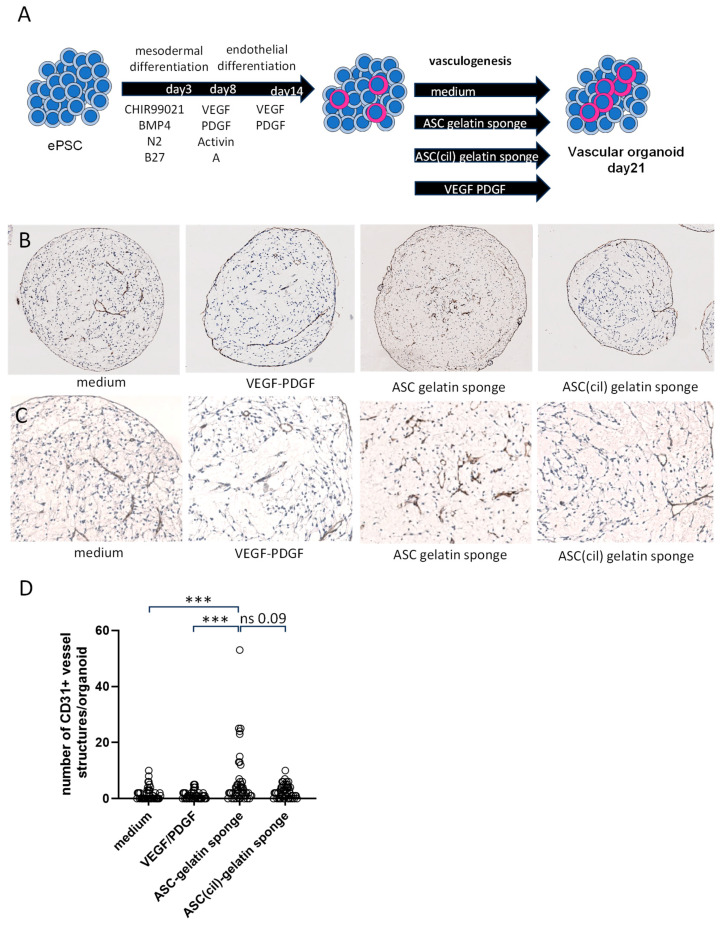
RGD motifs are involved in the secretion of factors from ASC in gelatin sponges influencing vasculogenesis in human vascular organoids. Media conditioned with ASC/gelatin sponge or ASC (cil)-gelatin sponge were compared for their capacity to influence the number of vascular structures in human vascular organoids derived from ePSC. (**A**) Experimental setup. (**B**) Low magnification of the CD31 staining of vascular organoids at day 21. (**C**) Higher magnification of the CD31 staining of vascular organoids at day 21. (**D**) Statistical analysis of 50 vascular organoids at day 21, from 2 independent experiments. *** *p* < 0.001; Mann–Whitney.

## Data Availability

Data supporting the reported results can be obtained upon request to the corresponding author of this study.

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
