# Peer review of "Adipose-Derived Stromal Cells Exposed to RGD Motifs Enter an Angiogenic Stage Regulating Endothelial Cells"

_ijms, 2025, doi:10.3390/ijms26030867_

Round 1
Reviewer 1 Report
Comments and Suggestions for Authors
This study provides valuable insights into the role of RGD motifs in enhancing the angiogenic potential of adipose-derived stromal cells (ASC) through integrin-mediated interactions. The manuscript is well-structured and addresses the complex interplay between inflammation and angiogenesis in tissue repair. The use of a gelatin matrix to mimic the wound environment and the application of a 3D vascular organoid model are notable strengths, providing a more physiologically relevant system for studying ASC behavior in angiogenesis and vasculogenesis. However, a few areas could benefit from further clarification and expansion:
-
While the study mentions upregulation of angiogenic factors (e.g., VEGF, HGF, and MMPs), specifying the extent of this upregulation or comparing it with physiological levels would enhance the understanding of ASC's response to RGD exposure.
-
The 3D vascular organoid model, while advantageous for studying human-specific tissue responses, is limited by its inability to replicate systemic interactions, full immune system dynamics, true vascular perfusion, and chronic or progressive conditions, which remain better studied in animal models. Please justify with due references.
-
While the manuscript highlights the therapeutic potential of ASC in tissue repair, a brief discussion on potential translational challenges, such as scalability or matrix compatibility in different tissue environments, would add depth to the conclusion.
-
Given the specific roles of α5β1 and αvβ3 integrins in angiogenesis, it would be beneficial to elaborate on how these integrins differentially contribute to the observed effects on endothelial cell behavior and vessel stabilization.
-
Including suggestions for future studies—such as investigating other ECM components that might work synergistically with RGD or testing additional integrin inhibitors—would provide a forward-looking perspective.
Overall, the study offers a solid foundation for understanding the role of RGD in ASC-mediated angiogenesis and its implications for regenerative medicine, but addressing these points could improve the manuscript’s clarity and impact.
Author Response
While the study mentions upregulation of angiogenic factors (e.g., VEGF, HGF, and MMPs), specifying the extent of this upregulation or comparing it with physiological levels would enhance the understanding of ASC's response to RGD exposure.
Thank you for this comment. This is a valuable suggestion to provide more explicit information on the extent of the upregulation of angiogenic factors. While physiological levels of these factors may not be suitable for comparison due to primarily local/paracrine secretion at healing sites without any known concentration, the extent of this upregulation can be better interpreted in terms of the nature, profile and significance of the increased factors/functions. A new paragraph has been added in the revised discussion section, discussing the extent of angiogenic regulations.
The 3D vascular organoid model, while advantageous for studying human-specific tissue responses, is limited by its inability to replicate systemic interactions, full immune system dynamics, true vascular perfusion, and chronic or progressive conditions, which remain better studied in animal models. Please justify with due references.
Yes, it is true that, although this model does not fully recapitulate the complete components and dynamics of angiogenesis in vivo, it offers the advantage of being a fully human model. To address this comment, a justification for the choice of this model and a brief discussion of its limitations, with reference, have been added to the discussion section.
While the manuscript highlights the therapeutic potential of ASC in tissue repair, a brief discussion on potential translational challenges, such as scalability or matrix compatibility in different tissue environments, would add depth to the conclusion.
That’s a useful and interesting comment. We have included these highlights in the discussion section of the revised manuscript.
Given the specific roles of α5β1 and αvβ3 integrins in angiogenesis, it would be beneficial to elaborate on how these integrins differentially contribute to the observed effects on endothelial cell behavior and vessel stabilization.
Yes, the existence of several RGD-binding integrins, including α5β1 and αvβ3, suggests distinct roles for these receptors. To investigate the specific roles of the three main RGD-binding integrins - α5β1, αvβ3, and αvβ5 - we tested additional inhibitors alongside cilengitide: volociximab (anti-α5β1), intetumumab (anti-αv), and etaracizumab (anti-αvβ3). The results were as follows: volociximab and etaracizumab did not prevent ASC attachment to gelatin, and intetumumab failed to reverse several of the endothelial functions studied. These findings indicate that the roles of RGD-binding integrins are highly complex, with probably significant compensatory effects between integrins. Additionally, the specificity of these inhibitors remains incompletely understood, further complicating data interpretation. For this reason, we opted to focus on the cyclic RGD inhibitor cilengitide, which broadly inhibits all integrins. Further studies, including experiments with knockout cell lines or animal models, will be necessary to precisely define the role of each integrin. To highlight this point, we have mentioned and discussed these unshown data in the revised discussion.
Including suggestions for future studies—such as investigating other ECM components that might work synergistically with RGD or testing additional integrin inhibitors—would provide a forward-looking perspective.
Yes, that’s a good suggestion. Regarding integrin inhibitors, please see our response to the previous point. Concerning other ECM components, we have included in the revised discussion - specifically in the final paragraph addressing scaling-up and therapeutic applications - a forward-looking perspective on the use of alternative RGD-delivering scaffolds.
Reviewer 2 Report
Comments and Suggestions for Authors
Dear authors,
I commend you for the effort and dedication evident in this important study on the interaction of ASCs with RGD motifs. My suggestions are intended to enhance the clarity and scientific rigor of your work.
Introduction
Comment. The introduction could further explore the specific gaps in the literature regarding ASCs and RGD. While the general role of ASCs in healing is described, it would be relevant to point out exactly what is still unknown about the influence of RGD on ASC functions and how the study will help fill these gaps.
Comment. The choice of the gelatin sponge to simulate RGD exposure is mentioned but not explained in depth. It would be helpful to discuss why this choice is appropriate and what advantages the gelatin model offers compared to other possible RGD and ASC study models, such as alternative scaffolds or 3D culture systems.
Comment. The introduction mentions the importance of angiogenesis and the role of ASCs but could discuss in more detail the stages of angiogenesis in which ASCs act (proliferation, migration, tubulogenesis) and how RGD specifically influences these stages, thus highlighting the study's contributions to understanding the angiogenic process.
Discussion
Comment. Discuss in greater depth the limitations of using gelatin sponges and vascular organoids and how these models may differ from in vivo systems. Recommend further studies, such as in vivo experiments or combined approaches that integrate other factors from the cellular microenvironment. Mentioning studies that explore these mechanisms would help ground this interpretation in established evidence.
Comment. Expand the discussion on potential clinical applications and the necessary steps to translate these findings into real therapeutic contexts.
Materials and Methods
Comment. To describe the experimental groups clearly, I suggest the authors add a specific section in the "Materials and Methods" titled "Experimental Groups." This section could detail each group used in both HUVEC assays and the vascular organoid model with hESCs, with a specification of the sample size (n) for each group.
ASC culture
Comment. Information on the age, sex, and health conditions of adipose tissue donors could be relevant, as these variables may influence ASC behavior. In addition, clarifying whether donors provided informed consent and if the study was approved by an ethics committee would strengthen the ethical transparency of the experiment.
The phrase mentions that "ASC lines were fully validated for their phenotype, multipotency, and regenerative potential," but does not specify the criteria or methods used for this validation. Describing cell surface markers (such as CD73, CD90, and CD105) and assays (such as differentiation into osteoblasts, adipocytes, and chondrocytes) would add greater clarity.
Comment. The final density of 6000 cells/mm³ is indicated, but a rationale for this concentration choice and the suspension volume used could clarify the density calculation. Explaining why this density was considered optimal helps strengthen the experimental design.
Comment. It would be helpful to specify how long the gelatin sponge with ASCs was maintained under air/liquid interface conditions and whether the culture medium was renewed during the experiment.
Comment. Information on which inhibitors were used, their concentrations, and a rationale for the 1-hour pre-incubation time is missing.
Migration, tubulogenesis, and permeability of endothelial cells
Comment. Insert the description of the cell type before the abbreviation "HUVEC" and indicate the source of these cells.
Generation of vascular organoids from embryonic pluripotent stem cells
Comment. It would be helpful to indicate the frequency of medium changes (if applicable) during the culture of organoids and whether the medium was renewed over the 21 days of culture. Specify the criteria used to evaluate organoid formation (e.g., size, morphology, presence of vascular markers).
Statistical analysis
Comment. Explain why the Mann–Whitney test was selected and whether normality was tested prior to the selection. The sample size of each group, if small, may limit the statistical power of the Mann–Whitney test. For a more robust study, the authors could ensure that the sample size was adequate to detect significant effects.
Comment. Since different comparisons were made (e.g., between groups with HUVECs and organoids under various conditions), a correction for multiple comparisons (e.g., Bonferroni or Holm-Bonferroni) could prevent false positives. If the correction was not applied, it may be relevant to include it to ensure the robustness of the statistical results.
Kind regards.
Author Response
The introduction could further explore the specific gaps in the literature regarding ASCs and RGD. While the general role of ASCs in healing is described, it would be relevant to point out exactly what is still unknown about the influence of RGD on ASC functions and how the study will help fill these gaps.
We thank the reviewer for this valuable comment. In the revised manuscript, we have improved the introduction and provided a more detailed presentation of the current knowledge regarding the gap between ASC and RGD presentation.
The choice of the gelatin sponge to simulate RGD exposure is mentioned but not explained in depth. It would be helpful to discuss why this choice is appropriate and what advantages the gelatin model offers compared to other possible RGD and ASC study models, such as alternative scaffolds or 3D culture systems.
Thank you for this valuable comment. In the revised introduction, we have provided a more detailed explanation of the rationale for selecting gelatin sponges as the RGD delivery method over other options. The key factors influencing this choice are as follows: (i) their known ability to deliver RGD motifs to cells more effectively than native collagen fibers, (ii) their capacity to mimic the denatured collagen fiber environment typically present in wound healing, (iii) their favorable physicochemical properties, such as porosity and flexibility, which facilitate the introduction of cells and culture medium, and (iv) their compatibility with clinical applications, including lack of toxicity, excellent biocompatibility, bioresorbability, and physicochemical characteristics that make them suitable for use in humans.
The introduction mentions the importance of angiogenesis and the role of ASCs but could discuss in more detail the stages of angiogenesis in which ASCs act (proliferation, migration, tubulogenesis) and how RGD specifically influences these stages, thus highlighting the study's contributions to understanding the angiogenic process.
Yes, based on the results presented in this study, it seems important to provide a more in-depth description in the introduction of the known impact of RGD-ASC interactions on components of angiogenesis. In response to the previous point requesting a more detailed explanation of the ASC/RGD interaction, we have also included a description, in the introduction, of the functional consequences of these interactions on key aspects of angiogenesis.
Discuss in greater depth the limitations of using gelatin sponges and vascular organoids and how these models may differ from in vivo systems. Recommend further studies, such as in vivo experiments or combined approaches that integrate other factors from the cellular microenvironment. Mentioning studies that explore these mechanisms would help ground this interpretation in established evidence.
This comment was also raised by another reviewer regarding the limitations of vascularized organoids. While it is true that this model does not fully replicate the complete components, particularly immune interactions, and dynamics of angiogenesis in vivo, it offers the distinct advantage of being a fully human-based system. To address this, we have included in the discussion section a justification for the choice of this model and a detailed discussion of its limitations, supported by references. Additionally, the revised manuscript now includes a discussion on the limitations of using gelatin sponges to mimic the wound healing environment (also in the discussion section). While gelatin sponges contain collagen fibers, they indeed lack the natural infiltration of inflammatory and angiogenic cells, which is a limitation of this model. Also, we have suggested further studies utilizing more integrative angiogenesis models, including in vivo approaches such as animal studies, as well as complementary in vitro assays.
Expand the discussion on potential clinical applications and the necessary steps to translate these findings into real therapeutic contexts.
This valuable comment was also raised by one reviewer. We have addressed it by expanding the discussion section to highlight the potential clinical translation of this research, with particular emphasis on scalability, the rationale for choosing the gelatin sponge for a therapeutic purpose, and its biocompatibility.
To describe the experimental groups clearly, I suggest the authors add a specific section in the "Materials and Methods" titled "Experimental Groups." This section could detail each group used in both HUVEC assays and the vascular organoid model with hESCs, with a specification of the sample size (n) for each group.
This section has been added in the revised manucript.
ASC culture : Information on the age, sex, and health conditions of adipose tissue donors could be relevant, as these variables may influence ASC behavior. In addition, clarifying whether donors provided informed consent and if the study was approved by an ethics committee would strengthen the ethical transparency of the experiment.
All ASC lines used were characterized for age, sex, health conditions. All were derived upon ethical approval. These informations have been added in the revised manucript (material and methods).
The phrase mentions that "ASC lines were fully validated for their phenotype, multipotency, and regenerative potential," but does not specify the criteria or methods used for this validation. Describing cell surface markers (such as CD73, CD90, and CD105) and assays (such as differentiation into osteoblasts, adipocytes, and chondrocytes) would add greater clarity.
These informations have been added in the revised manuscript.
The final density of 6000 cells/mm³ is indicated, but a rationale for this concentration choice and the suspension volume used could clarify the density calculation. Explaining why this density was considered optimal helps strengthen the experimental design.
This information has been added in the revised manuscript
It would be helpful to specify how long the gelatin sponge with ASCs was maintained under air/liquid interface conditions and whether the culture medium was renewed during the experiment.
This information has been added in the revised manuscript
Information on which inhibitors were used, their concentrations, and a rationale for the 1-hour pre-incubation time is missing.
This information has been added in the revised manuscript
Migration, tubulogenesis, and permeability of endothelial cells : Insert the description of the cell type before the abbreviation "HUVEC" and indicate the source of these cells.
This information has been added in the revised manuscript
Generation of vascular organoids from embryonic pluripotent stem cells : It would be helpful to indicate the frequency of medium changes (if applicable) during the culture of organoids and whether the medium was renewed over the 21 days of culture. Specify the criteria used to evaluate organoid formation (e.g., size, morphology, presence of vascular markers).
This information has been added in the revised manuscript
Statistical analysis: Explain why the Mann–Whitney test was selected and whether normality was tested prior to the selection. The sample size of each group, if small, may limit the statistical power of the Mann–Whitney test. For a more robust study, the authors could ensure that the sample size was adequate to detect significant effects. Since different comparisons were made (e.g., between groups with HUVECs and organoids under various conditions), a correction for multiple comparisons (e.g., Bonferroni or Holm-Bonferroni) could prevent false positives. If the correction was not applied, it may be relevant to include it to ensure the robustness of the statistical results.
The sample size/normality of distribution were checked to be adequate and justify the choice of the Mann-Whitney test. Justification was added, according to the following : A Mann-Whitney test was chosen to analyze the data as it is a non-parametric test suitable for comparing independent groups without a normal distribution.
Reviewer 3 Report
Comments and Suggestions for Authors
This study explores how RGD exposure influences ASC behavior, particularly in angiogenesis. The results show that ASC under RGD exposure elevated pro-inflammatory and pro-angiogenic factors and regulated certain biological behaviors of endothelial cells. Researchers also identified the crucial role of RGD-integrin interaction in the molecule's mechanism. These findings offer important views for the progression of cell-based regenerative therapies, yet there are still some points for improvement.
Major
1.Matrices similar to porcine gelatin sponge which contain RGD but without other potentially confounding factors could be contained in the experiment to better isolate and assess the specific effect of RGD.
2.The results of experiments that included different conditioned media should be described more specifically. Additionally, materials with similar compositions could be explored and details should be mentioned to enhance the scale of appliance guidance.
3.You mentioned that “the release of angiogenic factors further shifts the integrin expression of endothelial cells towards RGD-binding integrins” in the introduction. It would be beneficial to explore whether there are direct effects of RGD on endothelial cells, independent of ASC involvement.
4.Time-course experiments should be included, as you mentioned the dynamic regulation of angiogenesis in both the results and discussion. Considering temporal sequence would deepen the understanding of the factors and mechanisms involved.
5.In addition to comparing ASC performance in a gelatin sponge with and without cilengitide, adding a group of ASC in suspension could provide a more specific scope and extent of changes.
6.The exploration of RGD-binding integrin is insufficient, and more specific descriptions such as the classification and localization as well as relative experiments should be included.
7.The research used a single method for adding cilengitide and the rationale for this approach should be explained. If there is no special reason, alternative approaches, such as adding cilengitide to the sponge before introducing ASC, should be tested to broaden the scope of your findings.
8.Various inhibitors and different ways of inhibiting ASC-RGD interactions should be included to make the mechanism more comprehensive and convincing.
9.Figures demonstrating cell behaviors according to the statistics should be provided, for example, the tube forming of cells in Fig. 5. Besides, figures of ASC(cil) gelatin sponge group are missed in Fig.9B and Fig.9C.
10.The plastic material used to plate ASC for RGD-binding function analysis may differ significantly from gelatin sponge. The properties of the plastic should be explained, or alternative materials more similar to gelatin sponge should be considered.
11.Multiple methods of validation should be applied, such as analyzing specific gene expression through PCR.
12.The model mimicking could be more comprehensive by including animal experiments on tissue repair in damaged blood vessels, to demonstrate its therapeutic potential.
13.The potential of using RGD without a gelatin sponge and expanding its appliance scale can be explored and discussed.
Minor
1.The explanation of Figure B is unclear and the statistic annotations in the figure should be reviewed for accuracy.
2.The expression of “ASC/gelatin” in line 124 should be standardized in case of confusion.
3.The term “reverse” in line 187 should be reassessed after supplementing the experimental controls to confirm the degree of change.
4.The link between gelatin sponge, RGD, integrin, endothelial cells, and ASC as well as the mechanism of the related effects can be stated more logically in the introduction and discussion.
5.Line 149 “participate” should be “participates”
Author Response
1.Matrices similar to porcine gelatin sponge which contain RGD but without other potentially confounding factors could be contained in the experiment to better isolate and assess the specific effect of RGD.
Thank you for this insightful comment. We have indeed addressed the question of specificity controls by exploring the possible use of scaffolds that do not expose RGD motifs to ASC. Specifically, we tested the following conditions : collagen sponges (exposing few RGD, in a cryptic form compared to gelatin), collagen I/chondroitin-6-sulfate scaffolds, decellularized matrix from stomach sheep tissue, ASC clusters in 3D with the same ASC-to-medium ratio, and ASC spheroids with equivalent cell numbers. Our observations clearly indicated that each environmental condition created by the scaffolds uniquely regulated ASC behavior. This was evidenced by differences in the microtopography of ASC within the scaffold structures, as well as the influence of additional compounds that might interfere with biological responses. For instance, Principal Component Analysis (Figure 1) confirmed significant differences in the transcriptomes of ASC spheroids, ASC in gelatin sponges, and ASC in 3D culture without scaffolds. Given these unshown findings, we determined that using a scaffold without gelatin but with identical environmental parameters (shape, volume, porosity, pore size, etc.) was not scientifically feasible. Instead, the most appropriate specificity control for this study was the use of ASC cultured in gelatin sponges with RGD-binding integrins pre-blocked by Cilengitide. To address this crucial point regarding RGD specificity controls, we have added a dedicated paragraph to the Discussion section, providing further interpretation and justification.
2.The results of experiments that included different conditioned media should be described more specifically. Additionally, materials with similar compositions could be explored and details should be mentioned to enhance the scale of appliance guidance.
In response to this comment, we have clarified and more explicitly detailed the experiments testing the different conditioned media in the Results section. Additionally, to broaden the applicability of this study, we have included a discussion of materials with similar compositions in the revised manuscript. This has been addressed in a new paragraph in the Discussion section (as requested by reviewers 1 and 2), which explores in greater detail the challenges and considerations for scaling up and translating this work to clinical applications.
3.You mentioned that “the release of angiogenic factors further shifts the integrin expression of endothelial cells towards RGD-binding integrins” in the introduction. It would be beneficial to explore whether there are direct effects of RGD on endothelial cells, independent of ASC involvement.
We agree with this point. Indeed, while RGD motifs are primarily exposed to ASC by the "solid" scaffold, we cannot entirely rule out the possibility that some RGD-exposing gelatin fibers (collagen fibers) may become solubilized in the conditioned media and directly influence endothelial cell behavior independently of ASC. However, this scenario can be excluded in our experiments, as we included control conditions using empty gelatin sponges to account for the potential presence of soluble gelatin fibers exposing RGD. In line with this, we again recognize the importance of improving the Results section to better describe the experiments involving conditioned media.
4.Time-course experiments should be included, as you mentioned the dynamic regulation of angiogenesis in both the results and discussion. Considering temporal sequence would deepen the understanding of the factors and mechanisms involved.
We yet investigated the time course of angiogenic activation in ASC exposed to a gelatin sponge, analyzing the transcriptomes at 24 hours, 48 hours, 4 days, and 8 days. Our results showed that angiogenic regulation was established after 48 hours and subsequently maintained without significant changes between 48 hours and 8 days. Given that no specific additional regulations were observed during this period, but rather a sustained regulatory state, we propose to mention this experiment as "data not shown" in the Results section, as it is not critical to the overall findings of the study.
5.In addition to comparing ASC performance in a gelatin sponge with and without cilengitide, adding a group of ASC in suspension could provide a more specific scope and extent of changes.
In experiments analyzing the impact of conditioned media on HUVEC functions, it was not feasible to include the ASC monolayer condition. Indeed, in the 3D setups (ASC in gelatin sponge or ASC in 3D without a scaffold), the ASC-to-medium ratio was 1.5 × 10⁶ cells per 1 mL. This high ratio, favoring secretome concentration, was achieved through the clustering of ASC into a compact tissue structure under air-liquid interface conditions. In contrast, such a high ratio is unattainable in the context of an ASC monolayer in standard immersion, which explains why the compacted ASC without a scaffold was used as a control instead. This also highlights the advantages of using the gelatin sponge in combination with air-liquid interface conditions, as it provides an optimal experimental setup for studying ASC secretomes with enhanced sensitivity by concentrating their secreted factors.
6.The exploration of RGD-binding integrin is insufficient, and more specific descriptions such as the classification and localization as well as relative experiments should be included.
More description of RGD-binding integrins, according to this comment, has been introduced in the revised introduction.
7.The research used a single method for adding cilengitide and the rationale for this approach should be explained. If there is no special reason, alternative approaches, such as adding cilengitide to the sponge before introducing ASC, should be tested to broaden the scope of your findings.
We decided against pre-blocking the gelatin sponge with Cilengitide, opting instead to pre-block the ASC before introducing them into the gelatin sponge (after thorough rinsing). The rationale for this choice lies in the properties of gelatin as a widely used adsorbent for organic molecules. Gelatin is commonly employed in pharmaceutical applications as a matrix capable of adsorbing and progressively delivering organic compounds. Pre-exposing the gelatin sponge to Cilengitide would risk non-specific binding of the compound to the gelatin, thereby compromising the specificity of the experiments. To maintain experimental specificity, we chose to pre-block the ASC directly before rinsing and incorporating them into the gelatin sponge.
8.Various inhibitors and different ways of inhibiting ASC-RGD interactions should be included to make the mechanism more comprehensive and convincing.
Other reviewers have also raised this important and interesting point to be discussed. Yes, the existence of several RGD-binding integrins, including α5β1 and αvβ3, suggests distinct roles for these receptors. To investigate the specific roles of the three main RGD-binding integrins - α5β1, αvβ3, and αvβ5 – we have yet tested additional inhibitors alongside cilengitide : volociximab (anti-α5β1), intetumumab (anti-αv), and etaracizumab (anti-αvβ3). The results were as follows: volociximab and etaracizumab did not prevent ASC attachment to gelatin, and intetumumab failed to reverse several of the endothelial functions studied. These findings indicated that the roles of RGD-binding integrins were highly complex, with probably significant compensatory effects between receptors. Additionally, the specificity of these inhibitors remains incompletely understood, further complicating data interpretation. For this reason, we opted to focus on the cyclic RGD inhibitor cilengitide, which broadly inhibits all integrins. Further studies, including experiments with knockout cell lines or animal models, will be necessary to precisely define the role of each integrin. To highlight this point, we have mentioned and discussed these unshown data in the revised discussion.
9.Figures demonstrating cell behaviors according to the statistics should be provided, for example, the tube forming of cells in Fig. 5. Besides, figures of ASC(cil) gelatin sponge group are missed in Fig.9B and Fig.9C.
Thank you for this helpful comment. Figure 5 has been updated in the revised manuscript. Regarding Figure 9, we apologize for the copy/paste error; it has also been corrected in the revised version.
10.The plastic material used to plate ASC for RGD-binding function analysis may differ significantly from gelatin sponge. The properties of the plastic should be explained, or alternative materials more similar to gelatin sponge should be considered.
In fact, ASC were never cultured on plastic, further emphasizing the importance of using ASC clustered in 3D under air-liquid interface conditions rather than in monolayer cultures on plastic. To address and support this point, we suggest that Reviewer 3 refer to our response to Point 1.
11.Multiple methods of validation should be applied, such as analyzing specific gene expression through PCR.
We have proposed in Figure 7C a confirmation by PCR of the most significantly regulated genes identified by GSEA analysis as influenced by the gelatin sponge.
12.The model mimicking could be more comprehensive by including animal experiments on tissue repair in damaged blood vessels, to demonstrate its therapeutic potential.
To demonstrate the therapeutic potential of this research in vivo, we refer to the study by our group, Brembilla et al. (Biomedicines, 2023, Mar 22;11(3):987. doi: 10.3390/biomedicines11030987). While this previous published study did not specifically investigate RGD-induced activation pathways in ASC, it provides evidence that allogenic rat ASC in a gelatin sponge enhance angiogenesis in a rat model of ischemic wounds. This point has been included in the Discussion section of the revised manuscript.
13.The potential of using RGD without a gelatin sponge and expanding its appliance scale can be explored and discussed.
This point was also raised by other Reviewers, and we appreciate the suggestion. Regarding the scale of application, we have addressed this in the revised discussion, specifically in the final paragraph, where we include scaling-up and therapeutic applications as forward-looking perspectives.
Material and methods
1.The explanation of Figure B is unclear and the statistic annotations in the figure should be reviewed for accuracy.
The correction has been provided in the revised manuscript
2.The expression of “ASC/gelatin” in line 124 should be standardized in case of confusion.
The correction has been provided in the revised manuscript
3.The term “reverse” in line 187 should be reassessed after supplementing the experimental controls to confirm the degree of change.
The correction has been provided in the revised manuscript
4.The link between gelatin sponge, RGD, integrin, endothelial cells, and ASC as well as the mechanism of the related effects can be stated more logically in the introduction and discussion.
The correction has been incorporated into the revised manuscript, offering a more explicit and logical connection between these topics.
5.Line 149 “participate” should be “participates”
The correction has been provided in the revised manuscript
Reviewer 4 Report
Comments and Suggestions for Authors
The manuscript by Brembilla et al. has been focused on the pro-angiogenic properties of ADSCs and their influence on endothelial cells angiogenesis via RGD-dependent mechanism. In general, the study was well designed and the article is interesting. However, the description of methodology, results, and conclusions is not always clear and must be improved.
Below are my specific comments:
1. Methods - the Authors claim throughout the manuscript that the ADSCs were cultured on self-made porcine gelatin sponge, however on p. 17, line 421 there is a statement that cells were seeded on Spongostan sponge. Please, explain this inconsistency. Moreover, the Authors often write in methodological part "in some experiments", but it is not specified in which ones.
2. the Authors used "ADSCs in suspension" as a control for gelatin sponges, however, these cells are typically adherent cells not growing in suspension, so this term is misleading. Therefore, the control samples need more explanation on their growth mode and preparation.
3. Fig 1 B - please, explain what do the percentage numbers mean.
4. In Materials and Methods (ASC culture) - the Authors claim that ASC lines were fully validated - please, include te data as supplementary material.
5. The article is lacking "Conclusions".
6. page 16, line 413-414 - "...incubated at..." - please, complete the details properly
Author Response
- Methods - the Authors claim throughout the manuscript that the ADSCs were cultured on self-made porcine gelatin sponge, however on p. 17, line 421 there is a statement that cells were seeded on Spongostan sponge. Please, explain this inconsistency. Moreover, the Authors often write in methodological part "in some experiments", but it is not specified in which ones.
Thank you for this comment. It appears there may have been a misunderstanding, as, despite a thorough review of the manuscript, we did not find any mention suggesting that the gelatin sponge was self-made. It was, indeed, a commercially available, clinical-grade-compatible gelatin sponge, specifically Spongostan. Additionally, as requested, we have revised the manuscript to provide more details on the specific experiments referred to as "in some experiments.
- the Authors used "ADSCs in suspension" as a control for gelatin sponges, however, these cells are typically adherent cells not growing in suspension, so this term is misleading. Therefore, the control samples need more explanation on their growth mode and preparation.
Thank you for this comment, which we agree with. We acknowledge that the term "ASC in suspension" is unclear, as ASC typically grow as adherent cells. In response to this comment, we have replaced this term with "ASC suspension derived from a monolayer culture" in the revised manuscript. The question of controls has also been raised by other reviewers, and we agree that this is a pertinent suggestion. Accordingly, we have addressed the issue of specificity controls by investigating the potential use of scaffolds that do not expose RGD motifs to ASC. Specifically, we tested the following conditions : collagen sponges (exposing fewer RGD motifs, in a cryptic form compared to gelatin), collagen I/chondroitin-6-sulfate scaffolds, decellularized matrix from sheep stomach tissue, ASC clusters in 3D with an equivalent ASC-to-medium ratio, and ASC spheroids with similar cell numbers. Our findings demonstrated that each environmental condition provided by these scaffolds or setups uniquely regulated ASC behavior. This was evident from differences in ASC microtopography within the scaffold structures and the potential influence of additional compounds that might interfere with biological responses. For example, Principal Component Analysis (Figure 1) revealed significant differences in the transcriptomes of ASC spheroids, ASC in gelatin sponges, and ASC in 3D culture without scaffolds. Given these results, we concluded that using a scaffold without gelatin but with identical environmental parameters (shape, volume, porosity, pore size, etc.) was not scientifically feasible. Instead, the most appropriate specificity control for this study was to use ASC cultured in gelatin sponges with RGD-binding integrins pre-blocked by Cilengitide. To highlight this important point, we have added a dedicated paragraph to the Discussion section, providing further interpretation and justification.
- Fig 1 B - please, explain what do the percentage numbers mean.
The requested indication have been added in the figure legend
- In Materials and Methods (ASC culture) - the Authors claim that ASC lines were fully validated - please, include te data as supplementary material.
Thank you for this valuable comment. Another reviewer also requested information on the age, sex, and health conditions of the adipose tissue donors, as these variables may influence ASC behavior. All ASC lines used in this study were fully characterized regarding age, sex, and health conditions, and their derivation was conducted under ethical approval. As requested, this information has been included in the revised manuscript within the Materials and Methods section.
- The article is lacking "Conclusions".
That is a good suggestion. A conclusion has been added to the revised manuscript.
- page 16, line 413-414 - "...incubated at..." - please, complete the details properly
Thank you for the comment, the mistake has been corrected
Round 2
Reviewer 3 Report
Comments and Suggestions for Authors
I have no other suggestions.